# PrimKD: Primary Modality Guided Multimodal Fusion for RGB-D Semantic Segmentation

## ABSTRACT

The recent advancements in cross-modal transformers have demonstrated their superior performance in RGB-D segmentation tasks by effectively integrating information from both RGB and depth modalities. However, existing methods often overlook the varying levels of informative content present in each modality, treating them equally and using models of the same architecture. This oversight can potentially hinder segmentation performance, especially considering that RGB images typically contain significantly more information than depth images. To address this issue, we propose PrimKD, a knowledge distillation based approach that focuses on guided multimodal fusion, with an emphasis on leveraging the primary RGB modality. In our approach, we utilize a model trained exclusively on the RGB modality as the teacher, guiding the learning process of a student model that fuses both RGB and depth modalities. To prioritize information from the primary RGB modality while leveraging the depth modality, we incorporate primary focused feature reconstruction and a selective alignment scheme. This integration enhances the overall freature fusion, resulting in improved segmentation results. We evaluate our proposed method on the NYU Depth V2 and SUN-RGBD datasets, and the experimental results demonstrate the effectiveness of PrimKD. Specifically, our approach achieves mIoU scores of 57.8 and 52.5 on these two datasets, respectively, surpassing existing counterparts by 1.5 and 0.4 mIoU.

## CCS CONCEPTS

• **Computing methodologies** → *Image segmentation.*

## KEYWORDS

RGB-D Segmentation, Multimodal Fusion, Knowledge Distillation

## 1 INTRODUCTION

Semantic segmentation has been a long-standing research topic in the field of computer vision due to its wide application in autonomous driving and intelligent transportation systems. Recent advancements in modular sensors have paved the way for collecting multimodal data, presenting a promising opportunity to achieve significantly improved segmentation accuracy. For instance, RGB-D data, which combines RGB color channel information with depth information, has seen widespread use in semantic segmentation. By

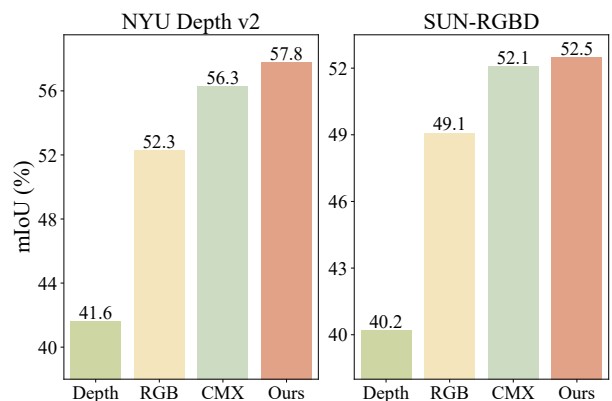

**Figure 1: Comparison between single modal based methods [26], an existing multimodal fusion approach CMX [30], and our proposed method on two datasets. Our primary modality guided fusion yields significant improvements in segmentation accuracy.**

integrating this additional depth information, RGB-D segmentation demonstrates significant improvements in distinguishing instances and context.

While the introduction of an additional modality does indeed offer more information, effectively harnessing and leveraging this additional information still poses a significant challenge. Since data from different modalities depict the scene in distinct manners, they cannot be processed separately as with a single modality. Therefore, achieving better multimodal fusion is significant for attaining remarkable performance.

Within contemporary research, dominant paradigms for information fusion in the context of multimodal segmentation can be categorized into two classes: static fusion approaches and interactive fusion approaches. In early static fusion methods, multimodal inputs are fused, after which a single network is employed to extract features from the fused inputs [3]. For example, Cao *et al.* [2] decomposes depth features into shape and base components, applies learnable weights to balance their importance, and performs convolution on the re-weighted combination to enhance segmentation accuracy. Alternatively, two backbones may perform feature extraction separately from the RGB and depth modalities, followed by fusion of these features for segmentation [4]. ACNet [9] combines RGB and depth features using attention complementary modules and a multi-branch architecture, allowing for selective feature gathering and fusion while preserving original RGB-D features for improved semantic segmentation performance. While these static fusion methods have achieved satisfactory performance, their straightforward approach often lacks inter-modal interactions, potentially resulting in inferior performance compared to single-modal methods [21].

Interactive fusion approaches have been proposed to enhance multimodal fusion. These approaches involve the integration of interactive fusion modules that link the RGB and depth features extracted by two parallel pretrained backbones, resulting in improved performance. For instance, TokenFusion [22] and HiDANet [24] dynamically fuse representations from RGB and depth encoders, aggregating them in the decoder, thereby pushing the performance boundaries in RGB-D scene parsing applications. Zhang *et al.* [30] also introduced a unified fusion framework, CMX, for RGB-X semantic segmentation. This framework incorporates a cross-modal feature rectification module for feature calibration and a feature fusion module for long-range context exchange and feature mixing, achieving superior segmentation performance. However, most existing interactive fusion approaches utilize identical backbone architectures to extract features from both RGB and depth data for fusion, overlooking potential differences between these two modalities. As the two modalities contain information of varying degrees, they may contribute differently to the final prediction. The use of identical backbones may suppress the effect of the primary modality, which contributes more to segmentation, while exaggerating the influence of the secondary one. Therefore, specialized designs are necessary to balance the participation of RGB and depth information for improved segmentation results.

To tackle this challenge, we introduce PrimKD, a knowledge distillation (KD)-based methodology, aimed at guiding multimodal fusion in RGB-D semantic segmentation, with a focus on leveraging the primary modality. Under the assumption of the varying importance between RGB and depth modalities in segmentation tasks, we employ a teacher model trained exclusively on data from the primary modality to guide the learning process of the fusion model. To enhance primary modality guided fusion, we employ both prediction-level and feature-level guidance strategies to attain superior results. Specifically, in addition to aligning final predictions, we explore direct reconstruction of intermediate features, integrating with an adaptive guidance scheme. In our experimental analyses, we empirically observe that employing an RGB data-trained teacher model leads to a notable enhancement in performance, thereby confirming the RGB data as the primary modality in RGB-D semantic segmentation. Extensive experiments conducted on two widely recognized datasets, namely NYU Depth V2 [19] and SUN-RGBD [20], validate the effectiveness of our proposed method. Specifically, our PrimKD achieves mIoU scores of 57.8 and 52.5 on these two datasets, respectively, outperforming the counterpart by a significant margin. Our contributions can be summarized as follows:

- We identify the discrepancy in the contributions of the RGB and depth modalities in RGB-D segmentation and propose a KD-based approach to enhance multimodal fusion, guided by the primary RGB modality, without altering the inference model architecture.
- We consider both prediction-level and feature-level guidance mechanisms for primary modality-guided fusion and validate an effective configuration that yields satisfactory multimodal fusion results.
- We conduct extensive experiments on two prominent RGB-D segmentation datasets to showcase the superiority of our proposed method, with ablation study demonstrating the effectiveness of each module within our framework.

## 2 RELATED WORKS

### 2.1 RGB-D semantic segmentation.

Semantic segmentation involves categorizing each pixel in an image into a specific class or object. Traditional approaches typically rely solely on RGB images for prediction. However, with the proliferation of depth sensors in recent years, the availability of depth data has increased significantly. This surge in data availability has led to a growing interest among researchers and practitioners in exploiting depth data to enhance RGB semantic segmentation. This emerging field is known as RGB-D semantic segmentation.

RGB images primarily capture color and texture details, whereas depth images predominantly convey spatial positional information. Integrating these modalities can enrich the semantic information of the original RGB images, leading to enhanced task performance to a certain extent. However, the disparity in information between these modalities presents a significant challenge in achieving effective fusion. To bridge this gap, various methods have been proposed, which can be categorized into two directions: developing dedicated feature extraction architectures tailored for RGB and depth image data [1–4, 6, 17, 28], and introducing innovative methods for feature alignment fusion [9, 18, 22, 23, 25, 31, 34, 35].

The first category of methods focuses on designing novel network architectures to better extract feature information from different modalities. For instance, Girdhar *et al.* [6] introduce a new Transformer architecture capable of jointly pre-training classification tasks across various modalities such as images, videos, and single-view 3D data, thereby achieving cross-modal semantic feature extraction. Zhang *et al.* [28] suggest incorporating depth images during pre-training to facilitate improved extraction of depth image features during fine-tuning.

The second category of methods involves designing innovative feature fusion approaches to align features from different modalities, thereby improving the semantic information of the RGB modality. For instance, Wu *et al.* [25] introduces a transformer-based fusion strategy to model context more effectively at the level of long-range information dependencies. Zhang *et al.* [30] propose cross-modal feature calibration modules and feature fusion modules to calibrate features of the current modality in both spatial and channel dimensions. They also utilize a cross-attention mechanism to globally enhance features from both modalities.

While existing methods have achieved notable performance enhancements through multimodal fusion, they often process information from different modalities using the same architecture. This can inadvertently lead to the suppression of the more informative modality, which we refer to as the primary modality, leading to a decrease in performance.

### 2.2 Knowledge Distillation.

Knowledge distillation (KD) is a technique used to transfer knowledge from one or more pretrained teacher models to a more compact student model. Depending on the type of information utilized for transferring, existing KD approaches can be broadly categorized into two groups: those that leverage the final predictions of the teacher and those that utilize intermediate features for distillation.

The vanilla KD method, as introduced by Hinton et al. [8], is the first to adopt the final predictions for KD. In this method, the output

distribution of the student classifier is required to mimic that of the teacher classifier by minimizing the KL divergence between the two distributions. Following the prior work, several enhancements to prediction-based KD have been proposed. Zhao *et al.* [32] introduced to decouple the original KL divergence into two separate parts and adjust the impact of these parts using hyperparameters. Huang *et al.* [10] focused on scenarios with large differences in model capacity between the teacher and student models and designed a Pearson correlation-based loss function. These improvements have demonstrated significant performance enhancements over the vanilla KD method.

In addition to using final predictions, intermediate features can also be leveraged for knowledge transfer. Romero *et al.* [15] proposed a method to directly align features of the teacher and the student using an additional convolutional layer for channel alignment. Apart from this straightforward design, various feature distillation approaches have been explored. For instance, attention maps [29], solution flow [5], and sample-level feature manifold representations [13, 14] are also viable knowledge formats. More recently, inspired by the success of masked image modeling [7], approaches introducing this idea into the KD field have emerged [27]. Moreover, integrating intermediate feature-based methods with final prediction-based approaches enables a more comprehensive amalgamation of information, thereby augmenting model performance.

Many existing KD approaches are tailored for model compression, prioritizing enhancement of the performance of lightweight student models. However, in this paper, we deviate from this norm by employing the teacher model as an external regularization factor to refine the multimodal fusion process of the student model.

## 3 METHOD

In this section, we provide an overview of existing RGB-D segmentation approaches, noting that many of them utilize identical architectures to process both modalities, inadvertently suppressing the primary modality containing more useful information. To address this limitation, we propose a KD-based approach. Our method leverages the primary modality to guide the multimodal fusion process, aiming to enhance segmentation performance.

### 3.1 Background

Semantic segmentation plays a crucial role in computer vision by assigning each pixel in an image to its corresponding semantic category. While models trained on RGB information excel in differentiating between colors and textures, they often struggle to capture geometric information, making it challenging to distinguish instances and contexts with similar visual characteristics [33]. To address this limitation, several studies have introduced depth images to enhance model performance [30, 31].

Existing methods commonly utilize two separate backbone models to extract features from both RGB and depth modalities. Given an input pair of RGB image $x_{\text{rgb}} \in \mathbb{R}^{3 \times H \times W}$ and depth image $x_{\text{depth}} \in \mathbb{R}^{H \times W}$, where $H$ and $W$ represent the height and width of the images, their features are extracted as follows:

$$f_{\text{rgb}} = \mathcal{M}(x_{\text{rgb}}, \theta_{\text{rgb}}),$$
$$f_{\text{depth}} = \mathcal{M}(x_{\text{depth}}, \theta_{\text{depth}}), \tag{1}$$

where $\mathcal{M}$ is the backbone model, with $\theta_{\text{rgb}}$ and $\theta_{\text{depth}}$ representing parameters of the corresponding backbones used for extracting features from the RGB and depth modalities, respectively. Subsequently, these features are fused together by a modal fusion module, which can be either statically or adaptively:

$$f_{\text{rgbd}} = \text{Fuse}(f_{\text{rgb}}, f_{\text{d}}). \tag{2}$$

Finally, a decoder module is employed to obtain the final prediction by taking $f_{\text{rgbd}}$ as input:

$$\hat{x} = \mathcal{D}(f_{\text{rgbd}}), \tag{3}$$

where $\mathcal{D}$ is the decoder module, $\hat{x} \in \mathbb{R}^{H \times W}$ represents the corresponding segmentation result of $x_{\text{rgb}}$.

### 3.2 Overlooked Modality Diversity

Integrating representations from both RGB and depth modalities has been shown to enhance model performance. As depicted in Equation 1, many current methods employ uniform architectures for feature extraction from each modality independently, differing only in parameters.

Since models with identical architectures share the same capacity, utilizing identical backbones implies treating both RGB and depth modalities uniformly. However, due to potential differences in information density between modalities, this approach may inadvertently suppress the primary modality containing richer information while exaggerating the secondary modality with less information, thus leading to suboptimal performance.

To address this disparity, specialized designs tailored to each modality are necessary. However, customizing model architecture for each modality can be cumbersome. Therefore, we aim to explore a simpler approach to calibrate the two modalities without modifying the model architecture, ensuring compatibility with existing RGB-D segmentation frameworks.

### 3.3 Primary Modality Guided Fusion

KD has been proven to be an effective approach for enhancing model performance by transferring knowledge from a pretrained teacher model to a student model. Drawing inspiration from its success, we aim to leverage a teacher model trained on the primary modality to guide the learning process of a fusion model incorporating both RGB and depth modalities. This approach allows us to achieve calibrated multimodal fusion.

Figure 2 presents an overview of our proposed primary modality guided multimodal fusion framework, termed PrimKD. Compared with existing multimodal fusion approaches that involve only an individual model learning from RGB-D data, our PrimKD framework incorporates an additional teacher model trained on the primary modality, which inherently contains more useful information for the segmentation task compared to the other modality. Building upon existing approaches that fuse multimodal information at the feature level, we further enhance the learning of the student model by employing KD guided by the teacher model. This is motivated by the assumption that the fused modality should resemble the primary modality more closely, thereby leading to improved performance. Our experiments demonstrate that RGB serves as the primary modality in RGB-D semantic segmentation.

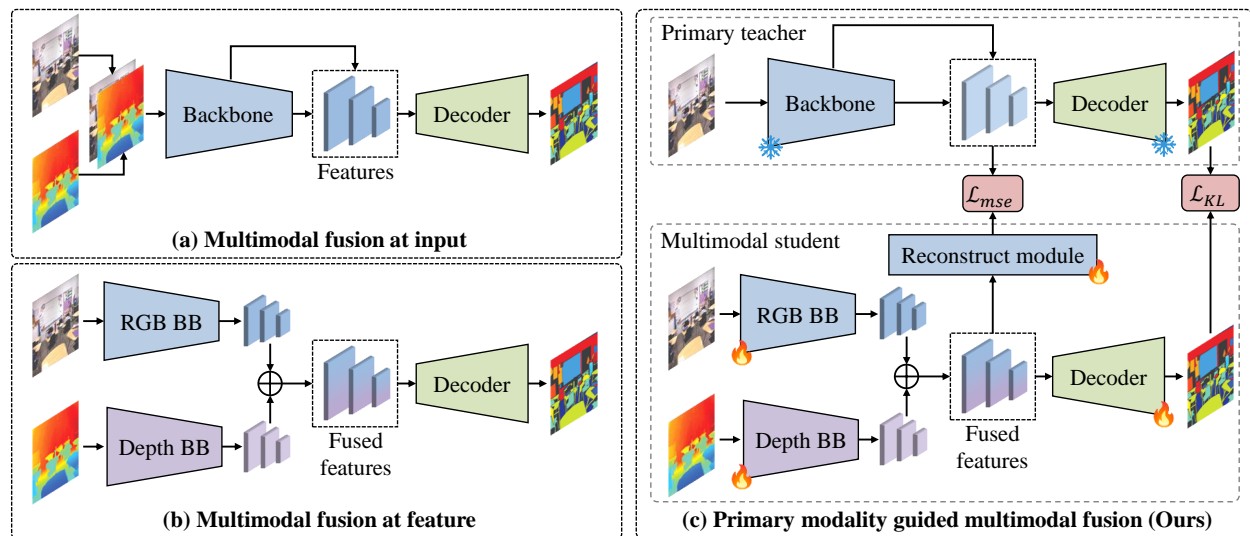

**Figure 2: Comparison of different multimodal fusion approaches for RGB-D semantic segmentation: (a) Multimodal fusion at input: The two modalities are fused and then processed by a single backbone to extract features for prediction. (b) Multimodal fusion at feature: Two individual backbones of the same architecture are used to extract features from each modality. These features are then fused for prediction. (c) Primary modality guided multimodal fusion (Ours): A teacher model trained on the primary modality, which contains more information, is used to distill a student model. Both prediction-level and feature-level KD are considered. This process calibrates the fusion by directing the student to pay more attention to the primary modality.**

*3.3.1 Prediction-level guidance.* Although the teacher and the student are trained with different data, they share the same output space. Therefore, we initially consider performing KD at the prediction level.

The teacher model is trained on the primary modality, its prediction process can be formulated as:

$$f_{\text{primary}} = \mathcal{M}_T(x_{\text{primary}}, \theta_T),$$
$$\hat{x}_T = \mathcal{D}_T(f_{\text{primary}}), \tag{4}$$

where subscript $T$ is used to indicate the teacher model. Additionally, we adopt $\hat{x}_S$ to represent the predictions of the student model, which is obtained following the process depicted in Section 3.1. Given that semantic segmentation involves classifying each pixel in the image to determine its corresponding class, we can apply KD at the prediction level, similar to traditional classification tasks. Specifically, the distillation loss is defined as following:

$$\mathcal{L}_{\text{pred}} = \mathcal{H}_{\text{KL}}(\psi(\hat{x}_T), \psi(\hat{x}_S)), \tag{5}$$

where $\mathcal{H}_{\text{KL}}$ denotes the KL divergence, and $\psi : \mathbb{R}^{H \times W} \to \mathbb{R}^{HW}$ represents an operation that flattens the segmentation result into a one-dimension vector.

Since the teacher model is trained exclusively using the primary modality, its predictions rely entirely on information from this modality. By training the student model to mimic these predictions, the fusion process is encouraged to prioritize information from the primary modality, resulting in improved fusion results.

*3.3.2 Feature-level guidance.* In addition to the final prediction, intermediate features have also been widely utilized in existing KD approaches. Therefore, we incorporate feature-level guidance

to transfer information from the primary modality. However, unlike in many existing studies where the teacher and student share the same model architecture, in our proposed PrimKD framework, the teacher model follows a single-stream architecture, while the student model adopts a dual-stream architecture. Considering our aim to impart more knowledge to the student model regarding the primary modality, it is reasonable to infer that the primary feature can be derived from the fused feature. Consequently, KD is performed using intermediate features of the teacher model and fused features of the student model. A lightweight module is employed to utilize fused features for reconstructing primary features, with the reconstruction error serving as the loss function. To explore the most suitable design, we consider several reconstruction schemes as follows:

*Direct reconstruction.* The simplest way to reconstruct the primary feature is to directly adopt a learnable module to take the fused features as input and output reconstructed features. This process can be formulated as:

$$\mathcal{L}_{\text{feat}} = ||\mathcal{R}(f_{\text{rgbd}}) - f_{\text{primary}}||_2, \tag{6}$$

where $\mathcal{R}$ is the learnable reconstruction module, which can consist of only a single layer or a sequence of layers. We evaluate different designs of this module in our experiments.

*Adaptive guidance.* Feature-level guidance involves comparing intermediate features from both the teacher and the student model. For semantic segmentation tasks, backbones typically consist of four stages, with each stage producing a feature map. These four feature maps of different scales are then utilized in the decoder module for prediction. Typically, reconstruction and comparison

entail the utilization of four feature maps. However, in our PrimKD scenario, the teacher and the student are trained using disparate datasets, resulting in divergence within their respective feature maps. Consequently, directly applying feature-level guidance across all four feature maps may present challenges.

To facilitate the knowledge transfer process, we explore two schemes. The first involves conducting feature-level guidance consistently on one of the four feature maps. The second scheme involves adaptive selection of features for reconstruction and comparison. Generally, this process can be formulated as follows:

$$\text{select rule}(\{\mathcal{L}_{\text{feat},i}|i \in \{1, 2, 3, 4\}\}), \tag{7}$$

where $\mathcal{L}_{\text{feat},i}$ represents the feature-level guidance loss associated with the $i$-th feature map. In our experiments, we will assess the impact of varying fixed map choices and employing different adaptive selection rules, including maximum- and minimum-based selection.

*Discussion.* Primary modality guided fusion mainly aims to utilize the primary modality trained teacher model as an external regularization to enhance the effectiveness of student learning of multimodal fusion. Particularly in feature-level guidance, the fused feature must encompass sufficient information from the primary modality for accurate reconstruction. Additionally, the segmentation target encourages the student to extract useful information from the secondary modality simultaneously. Ultimately, the student model is expected to achieve enhanced performance through a more balanced multimodal fusion.

## 3.4 Training and inference

The training of our PrimKD framework involves two stages. In the first stage, we train a teacher model with a single-stream architecture on the primary modality, identified as the RGB modality in our experiments. Subsequently, in the second stage, the student model is trained with the guidance of the frozen teacher. The total loss function during training is defined as:

$$\mathcal{L} = \mathcal{L}_{\text{seg}} + \alpha \mathcal{L}_{\text{pred}} + \beta \mathcal{L}_{\text{feat}}, \tag{8}$$

where $\mathcal{L}_{\text{seg}}$ represents the original RGB-D semantic segmentation loss, and $\alpha$ and $\beta$ are hyperparameters used to balance the two distillation losses.

The student model learns multimodal fusion strategy that taking more primary information into consideration. At inference time, the primary teacher model becomes unnecessary, resulting in no additional inference cost associated with our method.

## 4 EXPERIMENT

To assess the effectiveness of our proposed PrimKD method, we conduct experiments using two popular RGB-D segmentation datasets. Additionally, we conduct ablation studies to verify the effectiveness of each module within our framework. To initiate, we present our experimental setup.

## 4.1 Experimental setup

*4.1.1 Datasets.* In our experiments, we utilize two commonly referenced RGB-D segmentation datasets: NYU Depth V2 [19] and SUN-RGBD [20]. These datasets are frequently employed in existing literature [28, 30], and their details are outlined below:

**Table 1: Impact of different teachers on student performance with MiT-B4 backbone on NYU Depth V2. Results are obtained without multiscale testing. The RGB modality trained teacher achieves the highest student performance, showcasing RGB as the primary modality in RGB-D segmentation.**

| Teacher training modality | mIoU (%) |
|---|---|
| - | 56.1 |
| Depth | 56.2 |
| RGB | **57.5** |

*NYU Depth V2.* NYU Depth V2 comprises 1449 RGB-D samples covering 40 classes, with all RGB and depth images having a uniform resolution of 480×640. Among these, 795 image-depth pairs are allocated for training the RGB-D model, while the remaining 654 are reserved for testing.

*SUN-RGBD..* SUN-RGBD includes 10,335 RGB-D images with a resolution of 530×730, segmented into 37 categories. All samples in this dataset are divided into 5,285 segments for training and 5,050 segments for testing.

In both datasets, the depth images are converted into HHA format following [30]. The HHA format represents images with horizontal disparity, height above ground, and angle information.

*4.1.2 Models.* Given our primary goal of enhancing the multimodal fusion capability of existing dual-stream RGB-D semantic segmentation architectures, we opt not to introduce a novel model architecture. Instead, we adopt the dual-stream architecture proposed by Zhang *et al.* [30], which incorporates two MiT backbones[26]. In our experiments, we evaluate two different backbones: MiT-B2 and MiT-B4, both pre-trained on ImageNet-1K [16].

*4.1.3 Optimization.* Our training process consists of two stages. In the first stage, we train the teacher model using only the primary modality. Subsequently, in the second stage, we train the student model using both modalities with KD. Both stages share the same optimization configurations.

For the NYU Depth V2 dataset, we train models for 500 epochs with a batch size of 8. The initial learning rate is set to $6 \times 10^{-5}$ with a polynomial decay scheme. We employ AdamW [11] as the optimizer with a weight decay of 0.01. Data augmentation techniques including random flipping and random cropping are utilized. Experiments on the SUN-RGBD dataset follow a similar setting, with the only differences being the number of epochs, batch size, and initial learning rate, which are set to 300, 128, and $8 \times 10^{-5}$, respectively. The hyperparameters $\alpha$ and $\beta$ are set to 1 and 0.05, respectively, by default. All experiments are conducted using NVIDIA Tesla V100 GPUs. For the NYU Depth V2 dataset, we utilized 2 GPUs, while for the SUN-RGBD dataset, we employed 8 GPUs.

*4.1.4 Evaluation.* For evaluation, we employ the mean Intersection over Union (mIoU) metric. While many existing works typically report results obtained under multiscale testing, we employ this technique solely for comparison with these works. Our ablation experiments, in contrast, are conducted without multiscale testing.

**Table 2: Comparison of various multimodal fusion approaches for RGB-D segmentation, with all results obtained using multiscale testing. '-' indicates that corresponding results are not provided by the original paper.**

| Model | Backbone | Params | NYU Depth v2 | | | SUN-RGBD | | |
|---|---|---|---|---|---|---|---|---|
| | | | Input size | FLOPs | mIoU | Input size | FLOPs | mIoU |
| ACNet [9] | ResNet-50 | 116.6M | 480×640 | 126.7G | 48.3 | 530×730 | 163.9G | 48.1 |
| SGNet [3] | ResNet-101 | 64.7M | 480×640 | 108.5G | 51.1 | 530×730 | 151.5G | 48.6 |
| SA-Gate [4] | ResNet-101 | 110.9M | 480×640 | 193.7G | 52.4 | 530×730 | 250.1G | 49.4 |
| GEN [23] | ResNet-101 | 118.2M | 480×640 | 118.2G | 51.7 | 530×730 | 790.3G | 50.2 |
| ShapeConv [2] | ResNext-101 | 86.8M | 480×640 | 124.6G | 51.3 | 530×730 | 161.8G | 48.6 |
| ESANet [18] | ResNet-34 | 31.2M | 480×640 | 31.2G | 50.3 | 480×640 | 34.9G | 48.2 |
| TokenFusion [22] | MiT-B3 | 45.9M | 480×640 | 94.4G | 54.2 | - | - | - |
| TransD-Fusion [25] | Swin-B | 84.0M | 480×640 | - | 55.5 | 530×730 | - | 51.9 |
| Omnivore [6] | Swin-B | 95.7M | 480×640 | 95.7G | 54.0 | - | - | - |
| CMX [30] | MiT-B2 | 66.6M | 480×640 | 67.6G | 54.4 | 530×730 | 86.3G | 49.7 |
| CMX [30] | MiT-B4 | 139.9M | 480×640 | 134.3G | 56.3 | 530×730 | 173.8G | 52.1 |
| CMNext [31] | MiT-B4 | 119.6M | 480×640 | 131.9G | 56.9 | - | - | - |
| PrimKD (Ours) | MiT-B2 | 66.6M | 480×640 | 67.6G | 54.7 | 530×730 | 86.3G | 50.6 |
| PrimKD (Ours) | MiT-B4 | 139.9M | 480×640 | 134.3G | **57.8** | 530×730 | 173.8G | **52.5** |

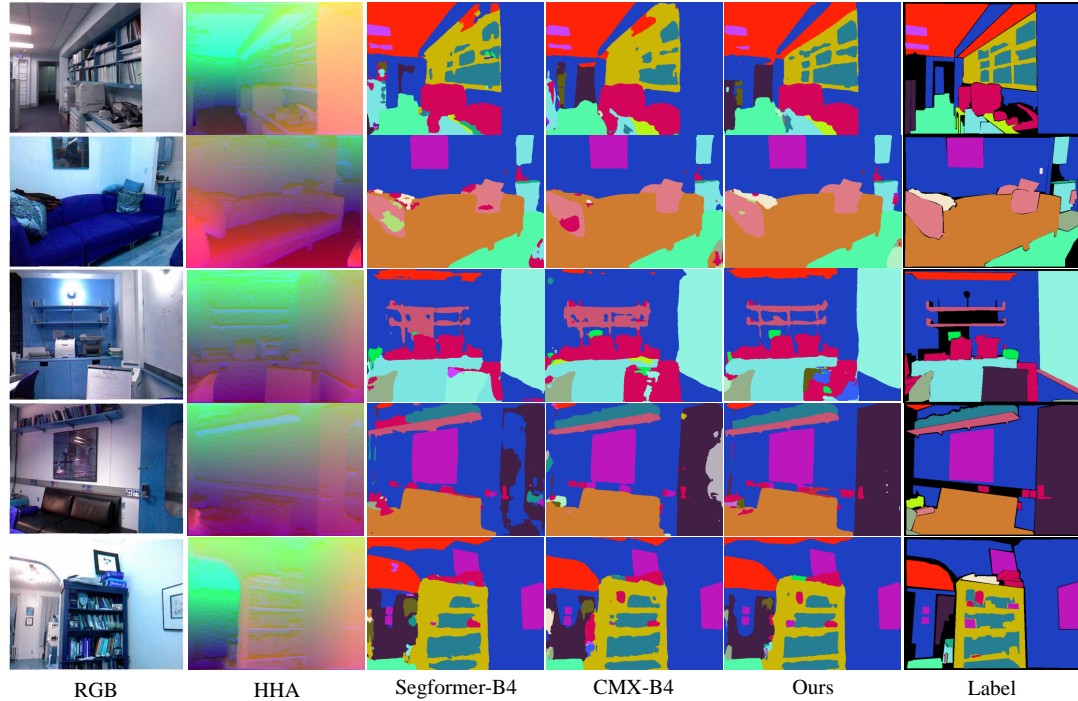

|  RGB | HHA | Segformer-B4 | CMX-B4 | Ours | Label |

**Figure 3: Visualization of segmentation results on the NYU Depth v2 dataset with the MiT-B4 backbone. Our proposed PrimKD method outperforms the single-stream method SegFormer [26] and the two-stream method CMX [30], which do not utilize primary modality guidance.**

## 4.2 Identification of the primary modality

In our analysis, we hypothesize that there exists a discrepancy in the significance of the RGB and depth modalities for the segmentation task, with one modality potentially providing more crucial information and thus serving as the primary modality. To ascertain the primary modality, we conduct experiments using teacher models trained on either modality to guide the learning of the student. We then compare these results with a non-KD baseline.

The comparison results are presented in Table 1. Without adopting any teacher, the student model achieves a test mIoU of 56.1 on the NYU Depth V2 dataset. When the model pretrained solely on the depth modality is adopted as the teacher, the student shows only a marginal improvement in performance, with a 0.1 mIoU increase. In contrast, when the RGB modality is used for teacher training, the student achieves a mIoU of 57.5, significantly outperforming its counterparts. These results demonstrate that constraining the fused features to contain more information from RGB data is beneficial. As a result, RGB is identified as the primary modality, and it will serve as the primary modality in all our subsequent experiments.

## 4.3 Primary Modality Guided Segmentation

To evaluate the effectiveness of our proposed method, we compare it with several recently proposed approaches on the NYU Depth v2 and SUN-RGBD datasets. Baseline results are obtained from [28]. Since our method primarily focuses on enhancing multimodal fusion performance for two-stream architectures, we do not include single-stream methods for comparison.

Table 2 presents the comparison results. On the NYU Depth v2 dataset, our PrimKD achieved an mIoU of 54.7 with the MiT-B2 backbone and 57.8 with the MiT-B4 backbone, surpassing the baseline CMX [30] without primary modality-guided multimodal fusion by 0.4 and 1.5 mIoU points, respectively. Similar performance improvements are observed on the SUN-RGBD dataset, with increases of 0.9 and 0.4 mIoU points using the MiT-B2 and MiT-B4 backbones, respectively. For clearer illustration of these improvements, we provide a bar plot in Figure 1. Furthermore, our method only incorporates the primary modality-trained teacher model during training, without modifying the architecture of the student model. This indicates that the performance improvements are achieved without sacrificing inference efficiency.

*Visualization.* To provide a qualitative comparison between our primary modality guided multimodal fusion method and existing approaches, we present visualization of several samples from NYU Depth v2 in Figure 3. From the results, model trained using our method is able to achieve more accurate segmentation results, further demonstrating the effectiveness of our design.

## 4.4 Ablation study

To further investigate the effectiveness of each module in our PrimKD framework, we conduct extensive ablation experiments. All ablation experiments are performed using the MiT-B4 backbone on the NYU Depth v2 dataset, with results reported without the use of multiscale testing.

*4.4.1 Prediction-level guidance and feature-level guidance.* We employ both prediction-level and feature-level guidance to direct the multimodal fusion process of the student model. To assess the effectiveness of each guidance method, we compare all possible combinations and present the corresponding results in Table 3. From the results, the introduction of either prediction-level or feature-level guidance improves performance, with enhancements of 0.5 and 0.4 mIoU, respectively. Furthermore, combining both forms of guidance yields the highest mIoU of 57.5, demonstrating the effectiveness of both approaches.

**Table 3: Impact of prediction-level and feature-level guidance on multimodal fusion results using the MiT-B4 backbone and the NYU Depth V2 dataset. Models trained with both types of guidance exhibit the best performance.**

| Prediction guidance | Feature guidance | mIoU (%) |
|:---:|:---:|:---:|
| × | × | 56.1 |
| ✓ | × | 56.6 |
| × | direct | 56.5 |
| ✓ | direct | **57.5** |

**Table 4: Assessing the effectiveness of adaptive guidance with the MiT-B4 backbone on the NYU Depth V2 dataset. Dynamically selecting the stage with the highest loss value for alignment yields the best performance.**

| Select rule | hint | mIoU (%) |
|:---:|:---:|:---:|
| × | stage 1 | 57.3 |
| × | stage 2 | 57.1 |
| × | stage 3 | 56.9 |
| × | stage 4 | 56.7 |
| average | stage 1-4 | 56.8 |
| minimum | stage 1-4 | 56.5 |
| maximum | stage 1-4 | **57.5** |

*4.4.2 Adaptive guidance.* In segmentation tasks, backbones typically output features of four different scales, referred to as features of stages 1 to 4. To identify which stage benefits the most from feature-level guidance and to evaluate the effectiveness of different adaptive stage selection rules, we conduct experiments and present the results in Table 4. When using features from a single stage, we observe a trend of decreasing performance as the model progresses to deeper stages, with the best performance achieved by utilizing features from stage 1 for guidance. Contrarily, averaging the loss values of all four stages does not lead to performance improvement. Moreover, adaptively selecting the stage with the maximum loss values for each sample yields the best performance, without the need for manual selection of where to apply guidance.

*4.4.3 Alignment architecture.* To reconstruct features of the teacher model, we utilize lightweight alignment modules. We examine different designs by comparing single-layer and multilayer architectures. The single-layer architecture consists of only a convolutional layer with a 1×1 kernel, while the multilayer architecture further incorporates a Conv-ReLU-Conv structure with a 3×3 kernels based on the single-layer aligner. The results, presented in Table 5, demonstrate superior performance of the multilayer architecture.

*4.4.4 Hyperparameters.* In our framework, there are two hyperparameters corresponding to the loss weight of prediction-level and feature-level guidance, respectively. We compare different configurations of these two hyperparameters and present the results in Table 6 and Table 7. From the results, the optimal combination of these two hyperparameters is $\alpha$ equal to 1 and $\beta$ equal to 0.05.

**Table 5: Comparison of alignment architecture designs with the MiT-B4 backbone on the NYU Depth V2 dataset. A multiple-layer architecture demonstrates superior performance compared to a single-layer design.**

| Architecture | Single layer | Multiple layers |
|---|---|---|
| mIoU (%) | 57.2 | **57.5** |

**Table 6: Comparison of different choices of hyperparameter $\alpha$ with the MiT-B4 backbone on the NYU Depth V2 dataset.**

| Weight of $\mathcal{L}_{\text{pred}}$ ($\alpha$) | 0.01 | 0.1 | 1 |
|---|---|---|---|
| mIoU (%) | 56.5 | 56.9 | **57.5** |

**Table 7: Comparison of different choices of hyperparameter $\beta$ with the MiT-B4 backbone on the NYU Depth V2 dataset.**

| Weight of $\mathcal{L}_{\text{feat}}$ ($\beta$) | 0.01 | 0.05 | 0.1 | 1 |
|---|---|---|---|---|
| mIoU (%) | 57.2 | **57.5** | 56.7 | 56.2 |

## 4.5 Analysis of guided fusion

To delve deeper into the impact of our proposed primary modality guided fusion, we analyze features of the trained fusion model. Specifically, we compare feature similarity using centered kernel alignment (CKA) [12] and feature value distribution through histogram analysis.

*CKA analysis.* To examine the feature similarity between dual modality trained models and single modality trained models, we utilize CKA as the measurement. CKA takes two feature maps as input and returns their similarity, ranging from 0 to 1, where a larger value indicates greater similarity.

We adopt the MiT-B4 as the backbone model and evaluate using the test set of NYU Depth V2. Table 8 presents the results. When guided fusion is not used, the features of the model trained using CMX achieve similar similarity compared to the single-stream model trained by either RGB or depth modality. Additionally, as the model goes deeper, the similarity also increases. After incorporating our primary modality guided fusion, the RGB modality trained teacher model is used to modulate the fusion of the student. Although the similarity to RGB modalities remains similar, there is a noticeable decrease in similarity of the trained model compared to the depth model, especially at the early stages of the model. This demonstrates that modulating the beginning stages is more significant, which is consistent with the result in Table 4.

*Feature value distribution analysis.* We further compare the distribution of intermediate feature values and present the histogram plot in Figure 4. The teacher model trained with the RGB modality extracts features with more concentrated values, while the CMX trained model without guidance presents scattered feature values. Equipped with our proposed method, the distribution of feature values shifts toward that of the teacher model, demonstrating that more information from the RGB modality is considered.

**Table 8: Comparison of feature similarity between our PrimKD trained fusion model and CMX with respect to RGB and depth modalities, using the CKA metric.**

| | stage 1 | stage 2 | stage 3 | stage 4 |
|---|---|---|---|---|
| | *Compare to RGB modality* | | | |
| CMX [30] | 0.60 | 0.63 | 0.71 | 0.91 |
| PrimKD (Ours) | 0.60 | 0.67 | 0.83 | 0.90 |
| | *Compare to Depth modality* | | | |
| CMX [30] | 0.44 | 0.50 | 0.61 | 0.88 |
| PrimKD (Ours) | 0.24 | 0.42 | 0.63 | 0.78 |

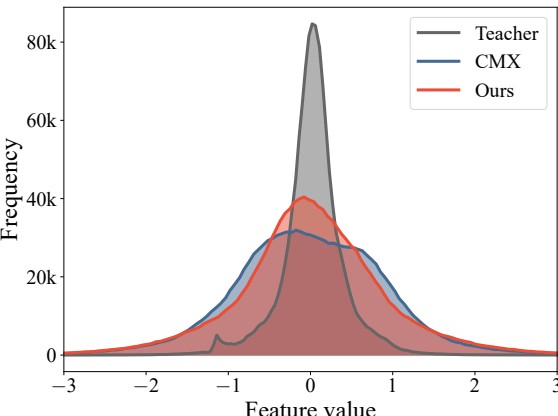

**Figure 4: Comparison of feature value distributions among models trained with CMX, our PrimKD method, and the RGB modality trained teacher model.**

## 5 CONCLUSION

Due to the recent advancements in depth sensors, the availability of depth images has significantly increased. Consequently, integrating these images with RGB images to enhance segmentation results has become increasingly appealing. Current methods often utilize two parallel backbones of the same architecture to extract features from each modality, followed by fusion modules for prediction. However, employing identical backbones overlooks the differences between these modalities, potentially resulting in suboptimal performance. To address this limitation, we introduce PrimKD, which employs a teacher model pretrained on the more informative modality to guide the multimodal fusion process of the student model under the KD framework. Both prediction-level and feature-level guidance are utilized in this approach. To enhance feature-level guidance, we further introduce adaptive guidance design. In our experiments, we identify that the RGB modality provides more information for the segmentation task, thus using it to train the teacher model. With our primary modality-guided multimodal fusion, the trained student model achieves an mIoU of 57.8 on the NYU Depth V2 dataset and 52.5 on the SUN-RGBD dataset, respectively, surpassing its counterparts by a significant margin while incurring no additional inference cost.

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
