# OpenReview forum: "PrimKD: Primary Modality Guided Multimodal Fusion for RGB-D Semantic Segmentation"
_acmmm.org/ACMMM/2024/Conference — MM2024 Poster_

### Official Review · Reviewer_3Jia · 2024-05-23

**Rating:** 4
**Confidence:** 2

**Summary:**

The paper introduces a novel approach leveraging knowledge distillation to enhance multimodal fusion in RGB-D semantic segmentation tasks. The core idea revolves around using a teacher model trained solely on RGB data to guide the fusion of RGB and depth data in a student model. This process aims to improve segmentation performance by ensuring that the fusion process prioritizes information from the RGB modality, which is identified as the primary modality containing more informative content for segmentation tasks.

**Strengths:**

- Novel Approach: The concept of using a knowledge distillation framework to prioritize the primary RGB modality in the fusion process is innovative. It addresses the common issue in existing fusion methods where equal weight is given to both RGB and depth modalities, which can dilute the more informative RGB signals.
&nbsp;
- Empirical Validation: The authors provide extensive experimental results to validate the effectiveness of the PrimKD method. The improvements are demonstrated on two standard datasets, NYU Depth V2 and SUN-RGBD, with the proposed method outperforming existing approaches in terms of mIoU scores.
&nbsp;
- Detailed Analysis: The paper includes a thorough ablation study that explores different aspects of the proposed method, such as the impact of prediction-level and feature-level guidance, the effectiveness of adaptive guidance, and the architecture of the alignment module. This comprehensive analysis strengthens the credibility of the results.
&nbsp;
- No Additional Inference Cost: It is advantageous that the primary modality-guided fusion introduced by PrimKD does not require the teacher model at inference time, which means there are no additional computational costs during the deployment phase.

**Limitations:**

- Complexity in Training: While the inference phase does not incur additional costs, the training phase could be significantly more complex and resource-intensive due to the dual-model setup and the need for knowledge distillation.
&nbsp;
- Lack of Broader Validation: The evaluation primarily focuses on RGB-D segmentation with specific datasets. Additional experiments across different modalities or segmentation tasks could help generalize the findings and prove the method’s applicability in other contexts.
&nbsp;
- Potential Overfitting: The method’s reliance on the primary RGB modality, while beneficial as shown, might also lead to overfitting to features predominant in RGB data, potentially ignoring useful cues from the depth data in some scenarios.
&nbsp;
- Limited Discussion on Failure Cases: The paper could benefit from a more detailed discussion on scenarios where PrimKD does not perform as expected, which would provide a deeper understanding of the method's limitations and scope.

**Suitability:**

3

---

### Official Review · Reviewer_sRSQ · 2024-05-24

**Rating:** 3
**Confidence:** 3

**Summary:**

This paper proposes a knowledge distillation method, called PrimKD, for improving the performance of RGB-D semantic segmentation. PrimKD focuses on leveraging a primary modality (RGB) during knowledge transfer to guide the learning of a student model that fuses both RGB and depth data. The authors consider both prediction-level and feature-level guidance mechanisms for primary modality-guided fusion.

**Strengths:**

The paper explores a KD-based method to improve the performance of multimodal semantic segmentation.

The experiment is conducted on two RGB-D datasets. The experiments show good results.

**Limitations:**

A key question is the technical novelty of PrimKD's knowledge distillation method. The paper needs to clearly explain how PrimKD's approach differs from existing work and what specific contributions it makes.

PrimKD utilizes an RGB-only model as the teacher. Exploring an alternative approach where a richer RGB-D model acts as the teacher could potentially improve student model performance.

The paper lacks details on the teacher and student model architectures. Additionally, discrepancies exist between PrimKD's parameter count and baseline methods like CMX. A comprehensive explanation of these architectural choices is necessary, particularly regarding their impact on both efficiency and performance.

The reported mIoU improvements achieved by PrimKD are modest compared to the baseline method without knowledge distillation. This raises concerns about PrimKD's overall effectiveness.

The evaluation focuses solely on segmentation performance. Including comparisons between different knowledge distillation methods would provide a broader context for PrimKD's effectiveness.

**Suitability:**

3

---

### Official Review · Reviewer_4a5C · 2024-05-29

**Rating:** 4
**Confidence:** 4

**Summary:**

The authors utilize a model trained exclusively on the RGB modality as the teacher, guiding the learning process of a student model that fuses both RGB and depth modalities.  The experiments are conducted on on the NYU Depth V2 and SUN-RGBD datasets, and the experimental results show the effectiveness.

**Strengths:**

It is  a KD-based way to enhance multimodal fusion, guided by the primary RGB modality. It considers both prediction-level and feature-level guidance mechanisms. The struture is simple, but achieves good multimodal fusion results.

**Limitations:**

(1) The fusion module in the manuscript is not clearly described.

(2) It is neccesary to discuss whether the fusion way can influence the effect of the KD learning.

(3) To be strict,  the mentioned adaptive guidance is not adaptive.

**Suitability:**

3

---

### Official Review · Reviewer_irth · 2024-05-30

**Rating:** 4
**Confidence:** 3

**Summary:**

The authors propose a new PrimKD method, which aims to leverage the knowledge learned from a model trained on the primary modality, to guide the learning process of a student model that fuses multiple modalities.

**Strengths:**

The authors discover that using identical backbones to extract information from multiply modalities will suppress the contribution of primary modality when multimodal fusion, and propose a KD-based method for enhencing the feature presentation of primary modality.

**Limitations:**

1.	The determination of the primary modal requires experimental verification in advance.  Besides, the autor consider the RGB is the primary modal in all experiments. How to consider different RGB-D application scenarios where RGB modal maybe not the primary modal?
2.	The improvement of the proposed method to CMX with MiT-B2 is 0.3 instead of 0.4 in 4.3.
3.	The visualization results between proposed and comparison methods shown in Figure 3 need to be eleborated in detail.
4.	Ablation experiments with two parameters in 4.4.4 need to explain in detail how one parameter is set when adjusting the other parameter.

**Suitability:**

3

---

### Meta-Review · Area_Chair_WDvB · 2024-07-01

**Recommendation:** Accept (Poster)
**Confidence:** 4

**Metareview:**

The paper introduces an approach using knowledge distillation to enhance multimodal fusion in RGB-D semantic segmentation tasks. The core idea revolves around using a teacher model trained solely on RGB data to guide the fusion of RGB and depth data in a student model. This process aims to improve segmentation performance by ensuring that the fusion process prioritizes information from the RGB modality, which is identified as the primary modality containing more informative content for segmentation tasks.

After the rebuttal, the paper receives three borderline accept and one borderline reject. The authors are encouraged to pay further attention to the limitation improvement mentioned by reviewer sRSQ.